# Chemophoresis engine: A general mechanism of ATPase-driven cargo transport

**Takeshi Sugawara**[1]*, **Kunihiko Kaneko**[2,3]

**1** Universal Biology Institute, The University of Tokyo, Tokyo, Japan, **2** Center for Complex Systems Biology, Universal Biology Institute, The University of Tokyo, Meguro-ku, Tokyo, Japan, **3** Niels Bohr Institute, University of Copenhagen, Copenhagen, Denmark

* take.sugawara@icloud.com

**Data Availability Statement:** All relevant data are within the manuscript and its Supporting information files.

**Funding:** This work was supported by Japan Society for the Promotion of Science (JSPS)

## Abstract

Cell polarity regulates the orientation of the cytoskeleton members that directs intracellular transport for cargo-like organelles, using chemical gradients sustained by ATP or GTP hydrolysis. However, how cargo transports are directly mediated by chemical gradients remains unknown. We previously proposed a physical mechanism that enables directed movement of cargos, referred to as chemophoresis. According to the mechanism, a cargo with reaction sites is subjected to a chemophoresis force in the direction of the increased concentration. Based on this, we introduce an extended model, the *chemophoresis engine*, as a general mechanism of cargo motion, which transforms chemical free energy into directed motion through the catalytic ATP hydrolysis. We applied the engine to plasmid motion in a ParABS system to demonstrate the self-organization system for directed plasmid movement and pattern dynamics of ParA-ATP concentration, thereby explaining plasmid equi-positioning and pole-to-pole oscillation observed in bacterial cells and *in vitro* experiments. We mathematically show the existence and stability of the plasmid-surfing pattern, which allows the cargo-directed motion through the symmetry-breaking transition of the ParA-ATP spatiotemporal pattern. We also quantitatively demonstrate that the chemophoresis engine can work even under *in vivo* conditions. Finally, we discuss the chemophoresis engine as one of the general mechanisms of hydrolysis-driven intracellular transport.

## Author summary

The formation of organelle/macromolecule patterns depending on chemical concentration under non-equilibrium conditions, first observed during macroscopic morphogenesis, has recently been observed at the intracellular level as well, and its relevance as intracellular morphogen has been demonstrated in the case of bacterial cell division. These studies have discussed how cargos maintain positional information provided by chemical concentration gradients/localization. However, how cargo transports are directly mediated by chemical gradients remains unknown. Based on the previously proposed mechanism of chemotaxis-like behavior of cargos (referred to as chemophoresis), we introduce a *chemophoresis engine* as a physicochemical mechanism of cargo motion, which transforms chemical free energy to directed motion. The engine is based on the

KAKENHI Grant No. 19H05796,17H06386, 17K15050 and Novo Nordisk Foundation. The funders had no role in study design, data collection and analysis, decision to publish, or preparation of the manuscript.

**Competing interests:** The authors have declared that no competing interests exist.

chemophoresis force to make cargoes move in the direction of the increasing ATPase (-ATP) concentration and an enhanced catalytic ATPase hydrolysis at the positions of the cargoes. Applying the engine to ATPase-driven movement of plasmid-DNAs in bacterial cells, we constructed a mathematical model to demonstrate the self-organization for directed plasmid motion and pattern dynamics of ATPase concentration, as is consistent with *in vitro* and *in vivo* experiments. We propose that this chemophoresis engine works as a general mechanism of hydrolysis-driven intracellular transport.

## Introduction

Cell polarity regulates the direction of intracellular transport for cargos, such as organelles and macromolecules, by taking advantage of chemical gradients sustained with the aid of ATP or GTP hydrolysis [1]. For example, it is well known that eukaryotic cell polarity factors, such as Rho, GTPase, and Cdc42, regulate the orientation of cytoskeleton members so that molecular motors can carry cargo directionally on the cytoskeleton, contributing to cell movement [2], cell growth [3], and axon guidance [4]. Although the transport by the cytoskeleton is one of the most commonly observed mechanisms, the transport directly mediated by chemical gradient, if its existence is confirmed, should be of importance as a general mechanism for cargo transport as well, which we refer to as cargo chemotaxis here.

A bacterial ParABS system [5–17] is a good candidate for cargo chemotaxis. It is the most ubiquitous bacterial polarity factor that regulates the separation of bacterial chromosome/plasmids into daughter cells by organizing their regular positioning along the cell axis [5–17]. Generally, it consists of three components as follows: The DNA binding protein ParB, ATPase, ParA, and the centromere-like site *parS*. ParB binds *parS*, spreads along the DNA, and forms a large partition complex (PC) around *parS*. ATP-bound ParA (ParA-ATP) can nonspecifically bind to DNA and interact with ParB-*parS* PC. Abundant ParA-ATP molecules are distributed on a nucleoid in a host cell. Their mobility is strongly restricted so that they are not homogenously distributed in the cell, thus enabling a sustained concentration gradient even within a micron-sized cell [13–15, 18–24]. Indeed, there are recent reports suggesting the existence of a concentration gradient in the *in vivo* experiments [25, 26]; They indicated that ParA-ATP gradient/localization can drive a *parS* site formation on a host genome/plasmid in the direction of the increased concentration, which can be a major candidate mechanism for plasmid partitioning and chromosome segregation [23–43].

ParA ATPase is an evolutionarily conserved protein which has many homologs [44–47]. Representative examples of its family are McdA/McdB ATPases controlling equidistribution of carboxysomes along a long cell axis in cyanobacteria [48–50], ParC/PpfA ATPases that regulate intracellular positions of chemotaxis protein clusters [51–53], MipZ ATPase that coordinates chromosome segregation in cell division [46, 54, 55], and MinD ATPase that determines a cell division plane [56–58]. These ATPase homologs, as well as ParA, work through a common mechanism essential to their function: Hydrolysis of an ATPase A by a partner protein B; A-ATP + B $\rightleftarrows$ C $\rightarrow$ A + ADP + B. By taking advantage of the free energy released by the reaction, a spatiotemporal pattern of the corresponding ATPase emerges [22, 46, 56–62], and cargo positions are coordinated [48–53]. One of the most renowned intracellular patterning systems is the MinCDE system that self-organizes the pole-to-pole oscillation of MinD, leading to the formation of a cell division plane at the cell center, upon stimulation of MinD ATPase activity induced by MinE at the inner cell membrane [57, 58]. In contrast, in the *in vitro* reconstitution of the Min system, traveling waves of MinD were observed [58–62].

Similar to the Min system, the pole-to-pole oscillation of ParA [14, 23, 63–67] also emerged in the ParABS system through the stimulation of ParA ATPase activity by ParB on the PC [13–15, 22–24]. Interestingly, a plasmid chases a ParA focus, following its oscillatory movement along the long host-cell axis [23], leading to oscillatory motion. In a recent *in vitro* experiment mimicking a ParABS system, Vecchiarelli et al. elegantly demonstrated the formation of directed motion of a cargo corresponding to a plasmid, referred to as "cargo surfing on ParA-ATP traveling wave" [28–30]. Hence, for both Min and Par systems, the emergence of traveling waves and the pole-to-pole oscillation of ATPase have been reported. The mechanism driving the plasmid motion, however, remains elusive [18], whereas the pattern dynamics of the Min system can be described by well-defined reaction-diffusion equations [58–60, 68–72].

Previously, we proposed a mechano-chemical coupling mechanism that enables directed movement of cargos, referred to as chemophoresis [1, 42, 43]. According to this mechanism, a macroscopic object with reaction sites on its surface is subjected to a thermodynamic force along an increasing concentration gradient. Cargo transport is possible via the chemophoresis force, [42, 43], and the possible role of the chemophoresis force in the separation dynamics of bacterial plasmids was discussed previously. By combining the plasmid motion driven by the chemophoresis force with a reaction-diffusion (RD) equation, we demonstrated that regular positioning of plasmids is possible in a ParABS system under ParA-ATP hydrolysis stimulated by ParB [16, 24, 64–67]. To date, however, spontaneous directed motion or pole-to-pole oscillation of plasmids [23, 64–67] has not been discussed in Ref [43], as was demonstrated by Vecchiarelli et al. [28–30] and theoretical studies [34–39].

In the present study, we extend our previous model and propose a *chemophoresis engine* as a general mechanism of cargo motion, which transforms chemical energy into directed motion via self-organization of the traveling wave, and then apply it to the plasmid motion in a ParABS system. In the previous study, the plasmid was assumed to be a point particle, where static equi-positioning and symmetric ParA-ATP distribution were robustly maintained [43]. However, such model with a zero-size limit is unrealistic, considering intracellular dynamics [26, 73] or reconstructing *in vitro* experiments performed by [28]. Here, by considering the finite size of plasmids explicitly, we show that organization of directed motion is possible via spontaneous symmetry breaking in the ParA-ATP pattern. We then recapitulate plasmid positioning to better describe the spatiotemporal profiles of ParA-ATP concentration and movement of plasmids. Actually, in the model presented here, the net chemophoresis force acts on the plasmid (PC) through the concentration difference between its ends, which is self-sustained by the high ATP hydrolysis stimulation. This self-driven mechanism leads to the directed motion of plasmids, as well as their equi-positioning [16, 24, 64, 67], and pole-to-pole oscillation as observed in bacterial cells and *in vitro* experiments [23, 64–67]. We mathematically show the existence and stability of the plasmid-surfing pattern, which allows cargo-directed motion through the symmetry-breaking transition of the ParA-ATP spatiotemporal pattern. We also indicate that plasmid size is a relevant parameter for the emergence of its directed movement. Finally, we quantitatively validate that the chemophoresis engine can work with parameters capturing *in vivo* conditions.

## Models

### Chemophoresis force

First, we briefly reviewed the chemophoresis force, a thermodynamic force acting on the cargo in the direction of the increased concentration of a chemical that can be bound on the cargo (See S1 Text for details.) We considered that a cargo was placed and moving in a $d$-dimensional space $\boldsymbol{r} \in \boldsymbol{R}^d$. The cargo had $N$ molecular sites B, on each of which $m$ molecules of

chemical X was bound to form a complex Y at position $r = \xi$. At each site, the reaction $mX(\xi)$ + B $\rightleftarrows$ Y occurred and was at chemical equilibrium. If a spatial gradient of chemical concentration X exists, the cargo is thermodynamically driven in the direction of the decreased free energy or the increased chemical potential of X [42, 43]. Here, such a gradient of the chemical potential $\mu(r)$ was assumed to be sustained externally through several active processes, supported by spatially distributed chemical gradients. We referred to the phenomenon as *chemophoresis*. The formula of the chemophoresis force was:

$$\boldsymbol{F} = mN_Y(\xi)\nabla\mu(\xi) = mNk_BT\frac{x(\xi)^m}{K_d{}^m + x(\xi)^m}\frac{\nabla x(\xi)}{x(\xi)} \tag{1}$$

Here, $\mu(r) = \bar{\mu} + k_BT \ln x(r)$, where $x(r)$ is the concentration of X, $K_d$ is the dissociation constant, and $m$ is the number of binding molecules corresponding to the Hill coefficient of the reaction. With this chemophoresis force, the cargo moved in the direction such that the concentration $x(r)$ increased even under thermal fluctuation ([42, 43], S1 Text). For the force to work, the reaction $mX + B \rightleftarrows Y$ was required to reach chemical equilibrium fast enough for cargo motion. Therefore, we showed that the chemophoresis force was one of the fundamental thermodynamic forces driven by physicochemical fields. Note that the force had an entropic origin from the viewpoint of statistical mechanics. See also Ref. [43] for details of the derivation from the viewpoint of thermodynamics and statistical mechanics.

To understand the origin of chemophoresis, it should be noted that microscopic binding events of X do not directly generate the force. Rather, the force works in the direction of larger frequency of the binding events (or larger time fraction of binding states) that was realized in the spatial location with a larger concentration of molecules in a chemical bath. The chemical gradient biases the binding frequency of X in a space-dependent manner. In other words, chemophoresis is driven by general thermodynamic force as a result of the free-energy (entropy) difference. It can also be derived by coarse-graining microscopic processes (S1 Text), whereas the macroscopic derivation implies its generality independent of specific microscopic models [35–37]. On the other hand, both the macroscopic (thermodynamic) and microscopic (statistical physics) theories are equivalent to each other, in that the force is generated with the aid of spatial asymmetry of molecule numbers bound on the bead, if its radius is finite. Further, for the chemophoresis force to act, X molecules do not necessarily have to bind cooperatively to the bead (as in the case of $m = 1$); if the concentration gradient of bound molecules is generated, the resultant free energy difference between its ends leads to the net chemophoresis force.

## Chemophoresis engine for plasmid partition

We then applied the chemophoresis formula to plasmid motion. As the reaction on the cargo consumed chemical X, its concentration changed; therefore, studied its RD equation. It was introduced for ParA-ATP ([42, 43], S1 Text), which recapitulates the central- and equi-positioning of plasmids [42, 43]. It was also adopted successfully to explain the directed movement of beads in an *in vitro* experiment by Vecchiarelli et al. [28]. We considered a plasmid $i(1 \leq i \leq M)$ placed into and moving in a $d$-dimensional space $r \in \boldsymbol{R}^d(d = 1$ or 2) (Fig 1A). ParA-ATP dimers were bound to a PC on plasmid $i$ at position $r = \xi_i$. $m$ ParA-ATP dimer molecules interacted with ParB, which stimulated ParA ATPase activity at a catalytic rate $k$ [74]; $N$ ParB molecules were assumed to be recruited to each PC at $r = \xi_i$. Because ParA could not bind PC when it was not combined with ATP, free ParA products were released from the PC immediately after ATP hydrolysis. Thus the reaction was presented as follows:

$$\mathrm{m(ParA - ATP)_2 + PC_i} \underset{k_-}{\overset{k_+}{\rightleftharpoons}} \mathrm{(ParA - ATP)_{2m} - PC_i} \overset{k}{\rightarrow} \mathrm{2mParA + 2mADP + PC_i} \tag{2}$$

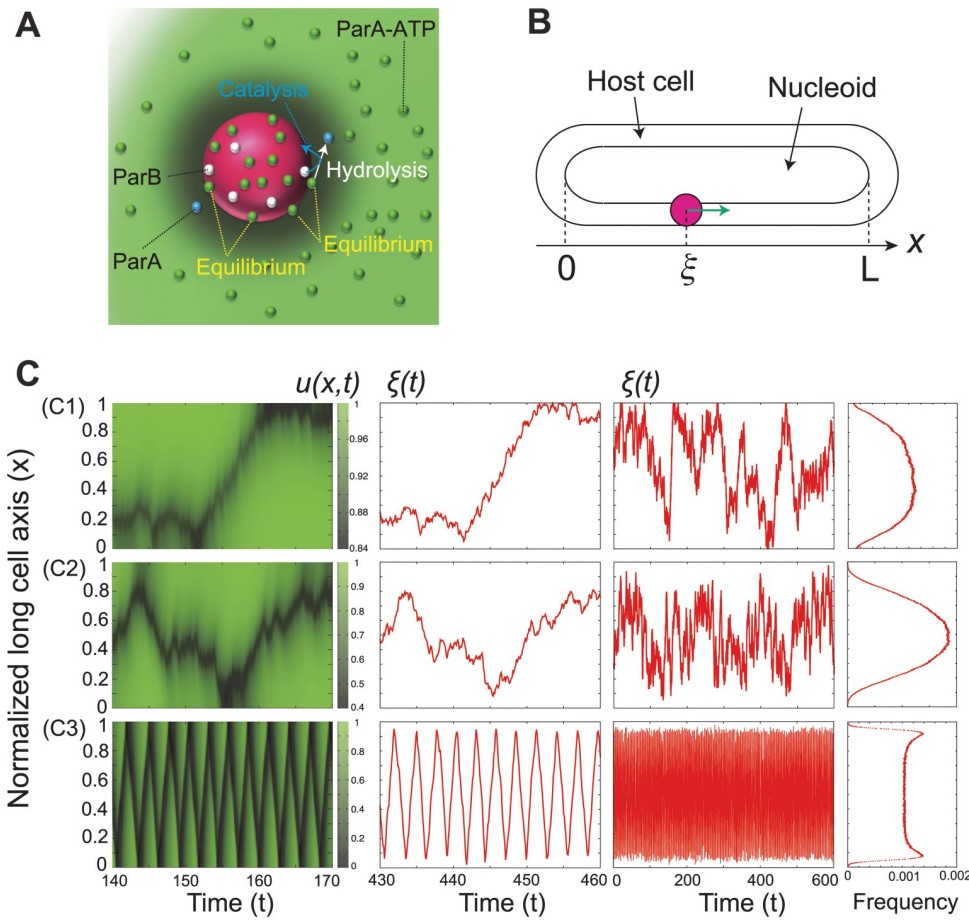

**Fig 1. Chemophoresis engine can recapitulate equi-positioning, directed movement, and pole-to-pole oscillation.**
(**A**) Schematic representation of the chemophoresis engine. A plasmid moves in a $d$-dimensional space $\boldsymbol{r} \in \boldsymbol{R}^d (d = 1$ or
2). ParA-ATP dimer (green sphere) binds a partition complex (PC, magenta sphere) on the plasmid at position $\boldsymbol{r} = \boldsymbol{\xi}_i$.
ParA-ATP dimer molecules interact with ParB molecules (white spheres), which stimulate ParA ATPase activity at a
catalytic rate. Because ParA cannot bind PC when it is not combined with ATP, free ParA products (blue sphere) are
released from the PC immediately after ATP hydrolysis. Through this reaction on PC$_i$ at $\boldsymbol{r} = \boldsymbol{\xi}_i$, each plasmid acts as a
sink for ParA-ATP and induces a concentration gradient of this protein. (**B**) One-dimensional case, on a nucleoid
matrix along the long cell axis where a plasmid $i(1 \le i \le M)$ is positioned at $x = \xi_i \in [0, L]$. (**C**) The dynamics change
among thermal motion, steady center-positioning, and directed movement followed by oscillatory mode as $\chi$ increases
among $\chi = 0.5$ (**C1**), $\chi = 2.5$ (**C2**), and $\chi = 10$ (**C3**) (two inner figures). (**C1**) The plasmid slightly tends to be localized
at the cell center but it is still dominated by thermal fluctuations for $M = 1$ and $\chi := kN/V = 0.5$. (**C2**) It is stably
localized at the cell center for $M = 1$ and $\chi = 2.5$, and (**C3**) it shows directed movement, reflection at the end walls, and
pole-to-pole oscillation for $M = 1$ and $\chi = 10$. The corresponding ParA-ATP pattern dynamics also change among
stochastic, steady center-positioning, and oscillatory waves (left). The oscillatory behavior of plasmids does not disrupt
time-averaged center-positioning, but steady center-positioning of plasmids are sustained (Compare (**C2**) right and
(**C3**), right). $K_d = 0.1$, $\varepsilon = 5$, and $L = 5$. The distributions (right) were generated using $10^7$ samples over $10^5$ time step.

Through this reaction on PC$_i$ at $\boldsymbol{r} = \boldsymbol{\xi}_i$, each plasmid acted as a sink for ParA-ATP and
induced a concentration gradient of this protein. In the early model, the size of plasmids was
assumed to be zero ([42, 43], S1 Text). However, the model is still too unphysical to better
reconstruct the movement of plasmids with a finite size in bacterial cells [26, 73] as well as that
of micro-sized beads in *in vitro* experiments [28]. To better describe spatiotemporal profiles of
ParA-ATP concentration and directed movement of the plasmids, we considered plasmids (or
PCs) as spheres with a radius of $l_b$, whose value is reported to be $l_b \sim 0.075$ $\mu$m in bacterial cells
according to [26, 73] and $l_b = 1.0$ $\mu$m in *in vitro* experiments [28].

Here, in order to discuss general situations, the derived equations were first rescaled by a dimensionless form and then numerical simulation was performed. Denoting the dimensionless concentration of ParA-ATP dimers on a nucleoid as $u(\boldsymbol{r})$, the normalized RD equation was written as follows (see S1 Text for its derivation):

$$\frac{\partial u(\boldsymbol{r})}{\partial t} = \nabla^2 u(\boldsymbol{r}) + (1 - u(\boldsymbol{r})) - \chi \frac{u(\boldsymbol{r})^m}{K_d^{\,m} + u(\boldsymbol{r})^m} \sum_{i=1}^{M} \theta(l_b - |\boldsymbol{r} - \boldsymbol{\xi}_i|) \tag{3}$$

where the first and second terms represent the diffusion of ParA-ATP and its chemical exchange at a normalized constant rate with the cytoplasmic reservoir (denoted by its normalized concentration), respectively. The last term denotes the inhibition by ParB on the $M$ PCs. $K_d$ is the normalized dissociation constant of the reaction $mX + B \rightleftarrows Y$, and $m$ is the Hill coefficient. $V$ is the $d$-dimensional volume of the bead with a radius of $l_b$. $\chi = kN/V$ is a maximum rate for ParA-ATP hydrolysis by ParB on each PC (S1 Text). Furthermore, $\theta(\boldsymbol{r})$ is a step function representing the space each PC occupies to describe the hydrolysis reaction space. Only within $|\boldsymbol{r} - \boldsymbol{\xi}_i| < l_b$, the reaction occurred. Without the last term (if $\chi = 0$), $u(\boldsymbol{r})$ reached a homogenous equilibrium state, $u(\boldsymbol{r}) = 1$. In contrast, the normalized equations of motion for plasmids were represented as follows:

$$\frac{d\boldsymbol{\xi}_i}{dt} = \varepsilon \int d\boldsymbol{r} \frac{u(\boldsymbol{r})^m}{K_d^{\,m} + u(\boldsymbol{r})^m} \frac{\nabla u(\boldsymbol{r})}{u(\boldsymbol{r})} \theta(l_b - |\boldsymbol{r} - \boldsymbol{\xi}_i|) + \boldsymbol{\eta}_i(t) \tag{4}$$

with thermal noise $\langle \boldsymbol{\eta}_i(t) \rangle = 0$ and $\langle \boldsymbol{\eta}_i(t) \cdot \boldsymbol{\eta}_j(t') \rangle = 2d\mathcal{D}\delta_{ij}\delta(t - t')$, and $\varepsilon = \mathcal{D}N/V$. Here, $\mathcal{D} = D_\xi/D_u$ is the relative diffusion coefficient of the plasmid to that of ParA-ATP (see S1 Text for details). The parameters to be assigned to Eqs 3 and 4 are $K_d, l_b, m, M, N, \mathcal{D}, k$, and the system size $L (=$ cell length).

## Results

### Chemophoresis engine captures observed plasmid dynamics

**Chemophoresis engine can recapitulate equi-positioning, directed movement, and pole-to-pole oscillation.** First, we considered the motion in a one-dimensional (1D) space ($d = 1$), that is, on a nucleoid matrix along the long cell axis where a plasmid $i(1 \leq i \leq M)$ was positioned at $x = \xi_i \in [0, L]$ (Fig 1B); The Neumann boundary condition was adopted for the RD equation: $\nabla u(0) = \nabla u(L) = 0$. To confine the plasmids to the host cell $x \in [0, L]$, we placed the reflection walls at $x = 0$ and $x = L$. This could be explicitly represented as $U_b = \begin{cases} 0 & 0 < x < L \\ \infty & \text{otherwise} \end{cases}$. We set $l_b = 0.2$, $V = 2l_b = 0.4$, $m = 1$, $N = 40$ through the simulation, and then we examined how the plasmid dynamics change with $\chi := kN/V$, where $\chi$ is the normalized maximum rate for ParA-ATP hydrolysis by ParB on each PC. For $M = 1$, the dynamics of $u(x)$ and the plasmid, as well as the distribution of the plasmid position, are displayed for $\chi = 0.5, 2.5$ and $10$ in Fig 1C. For $\chi = 0.5$, the plasmid slightly tends to be localized at the cell center, but it is still dominated by thermal fluctuations (Fig 1C1). The plasmid was stably localized at the cell center for $\chi = 2.5$ (Fig 1C2), whereas for $\chi = 10$, it showed directed movement and then reflected at the end walls, resulting in pole-to-pole oscillation (Fig 1C3). In general, the plasmid showed directed motion for a larger $\chi$ (= maximum rate of ParA-ATP hydrolysis). This result was plausible because the larger $\chi$ generates the sharper gradient of ParA-ATP, leading to the larger chemophoresis force to enable the persistent directed motion of the plasmid.

Similarly, for $M > 1$ cases, plasmid dynamics qualitatively changed among stochastic switching, steady equi-positioning, and directed movement followed by an oscillatory mode as

$\chi$ increased (S1 and S2 Figs). We then examined how plasmid dynamics switched from a static to an oscillatory mode with increasing $\chi$. The switch in plasmid dynamics occurred through a symmetry-breaking transition of the ParA-ATP spatiotemporal pattern. Interestingly, the oscillatory behavior of plasmids did not disrupt time-averaged equi-positioning. Steady multimodal distribution of plasmids was sustained (S1 and S2 Figs). As reported previously [42, 43], the regular positioning of plasmids is due to the effective inter-plasmid repulsive interaction derived from the chemophoresis force. The plasmid acting as a sink for ParA-ATP contributed to the formation of a concentration gradient, which increased with the distance from the plasmid. Other plasmids were subjected to the chemophoresis force caused by the gradient in the direction of increasing ParA-ATP concentration so that they were forced away from the former. The former plasmid was also subjected to the chemophoresis force caused by the gradient derived from the latter, resulting in mutual repulsion among the plasmids. The mutual repulsive interaction contributed to the robustness of the positional information generated by the chemophoresis engine. This plasmid separation scenario by such repulsive interactions is consistent with a previous observation [16].

**Chemophoresis engine mathematically validates plasmid surfing on the traveling ParA-ATP wave.** In a recent report, [28], Vecchiarelli et al. demonstrated the directed movement of micro-sized beads that mimic plasmids. A theoretical explanation using more realistic model with finit-sized plasmids is needed, as the previous models [43] assumed vanishing-size plasmids; Hence, we introduced the $l_b$-sized plasmids in Eqs 3 and 4, to discuss a possible symmetry-breaking transition more realistically. Here, we analytically examined plasmid surfing on the ParA-ATP traveling wave, focusing only on $M = 1$, without thermal fluctuation, for a 1D case under a periodic boundary condition.

S3 Fig shows simulation results of Eqs 3 and 4 for $\chi = 2.5$ (S3A Fig) and $\chi = 10$ (S3B Fig) with $K_d = 0.001$. In the former case, the plasmid maintained its location, whereas in the latter it traveled on the $u(x)$ wave and moved unidirectionally. Fig 2A shows a steady velocity ($= v$) profile of plasmid movement for $0 < \chi(= kN/(2l_b)) < 10$ and $0 < \varepsilon(= \mathcal{D}N/(2l_b)) < 10$. In the case without thermal fluctuation, as $\chi$ and $\varepsilon$ increased, the velocity monotonously increased above a critical curve on the $\chi - \varepsilon$ plane (Fig 2A). Above the curve, the plasmid showed directed motion.

To analytically examine the change in the steady solutions of plasmid dynamics against $\chi$ values, we simplified Eqs 3 and 4 assuming $u(x) \gg K_d$ over $x \in [0, L]$ resulting in $\frac{u(x)^m}{(K_d^m + u(x)^m)} \to 1$ (S2 Text). Furthermore, by introducing a co-moving frame with a space-time coordinate $z :=$ $x - vt$, where $v$ is the steady velocity of the plasmid, defining $u(x, t) := U(x - vt, t)$, and solving the steady-state equation S3 and S4 Eqs in S2 Text, we obtained a relationship between $v$ and $\chi$ (Fig 2B, green solid line) and the steady-state solution $U^{st}(z)$ (Fig 2C, green solid line) (S2 Text). Solutions for directed movement ($|v| > 0$) emerged at $\chi = \chi_c \sim 3.1$ as a result of a (supercritical) pitchfork bifurcation, whereas the localized solution without motion ($v = 0$) existed over $0 \le \chi \le 10$. Hence, there were three solutions for $\chi > \chi_c$ (S2 Text and S4 Fig). Fig 2C shows steady solutions $U^{st}(z)$ for localization (purple) at $\chi = 2.5$ and directed movement (green) at $\chi = 10$. Note that the plasmid location is fixed at the origin ($z = 0$) on the space-time coordinates (S2 Text). For the case of $v = 0$, the shape of $U^{st}(z)$ was symmetrical and had its minimum at the origin, reaching an equilibrium state of the plasmid location (Fig 2C, purple). In contrast, the symmetry of $U^{st}(z)$ was broken for the case of $|v| > 0$, supporting a non-equilibrium traveling wave (Fig 2C, green). Interestingly, the minimum of the latter traveling wave was positioned at a location shifted from the origin of the plasmid (Fig 2C, inset figure, green), suggesting that the plasmid was "surfing" on the traveling wave. Next, we numerically calculated the steady velocity using Eqs 3 and 4 with $K_d = 0.001$ over $0 \le \chi \le 10$ and confirmed the

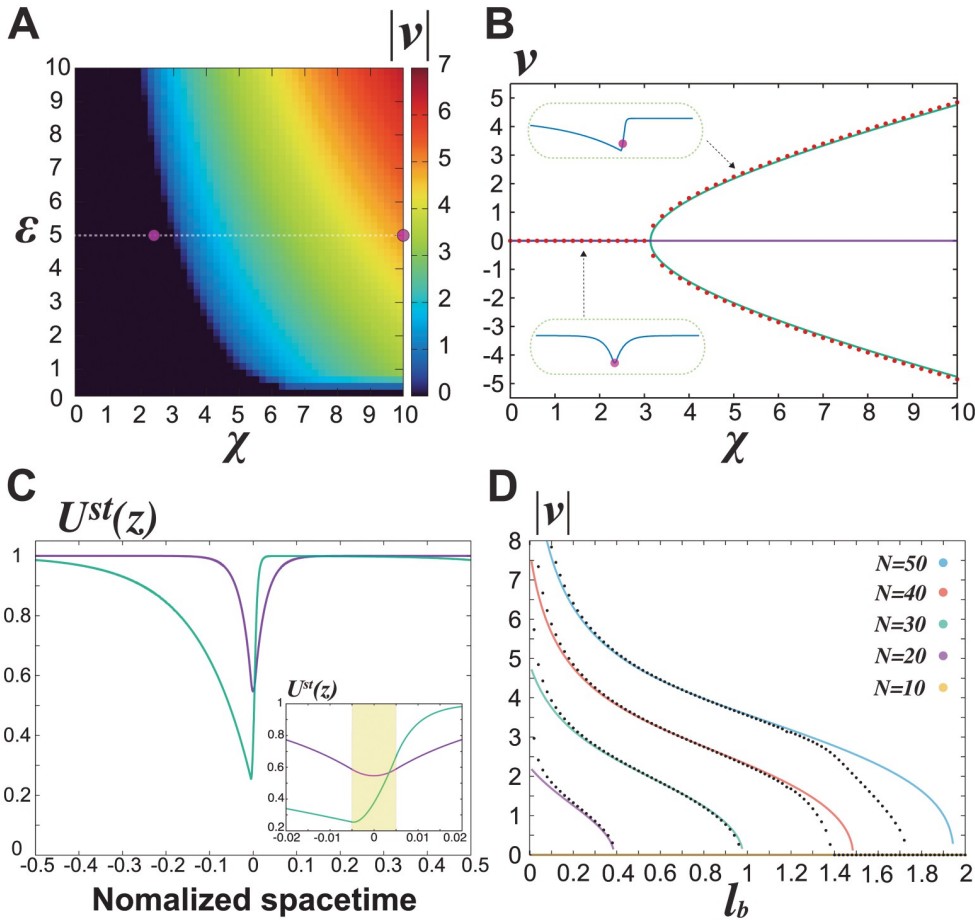

**Fig 2. Chemophoresis engine mathematically validates plasmid surfing on the traveling wave of ParA-ATP. (A)** Steady velocity ($|v|$) profile of plasmid movement for $0 < \chi < 10$ and $0 < \epsilon < 10$ without thermal fluctuations. The plasmid starts moving above a critical curve on the $\chi - \epsilon$ plane. The white dotted line shows a parameter region in Fig 2B. The magenta dots at $(\chi, \varepsilon) = (2.5, 5)$, $(10, 5)$ corresponds to parameter values for the steady solutions shown in Fig 2C. $K_d = 0.001$, $N = 40$, and $L = 40$. **(B)** Relationship between $v$ and $\chi$ in analytical (green for $v \neq 0$ and purple solid line for $v = 0$) for S3 and S4 Eqs in S2 Text, and simulated (red dots) solutions for Eqs 3 and 4. Solutions for directed movement ($|v| > 0$) emerge at $\chi = \chi_c \sim 3.1$ as a result of a supercritical pitchfork bifurcation, whereas a solution for localization ($v = 0$) exists over $0 \leq \chi \leq 10$. $K_d = 0.001$, $N = 40$, and $L = 40$. **(C)** Analytical solutions of the ParA-ATP pattern $U^{st}(z)$ in localization (purple) at $\chi = 2.5$ and directed movement (green) at $\chi = 10$ for S6 and S7 Eqs in S2 Text. The inset figure shows an enlarged view of $U^{st}(z)$ for $z/L \in [-0.02 : 0.02]$. The plasmid location is fixed at the origin ($z = 0$) on the space-time coordinates. $K_d = 0.001$, $N = 40$, and $L = 40$. **(D)** $l_b$ dependency of the directed motion of the plasmid for the analytical solution S7-S9 Eqs in S2 Text (solid lines) and numerical result calculated from Eqs 3 and 4 (black dots). For $N = 20, 30, 40, 50$, there exists a solution with the directed movement, whose velocity monotonously decreased with $l_b$, and an inverse pitchfork bifurcation occurred at a critical value of $l_b$, resulting in the only solution with $v = 0$. The numerical results showed slight deviation from the analytical solution for the ranges of small and large $l_b$, suggesting the breakdown of the approximation $u(x) \gg K_d$. For clarity, the numerical result for $v = 0$ was displayed only in the case of $N = 40$. $k = 0.1, \mathcal{D} = 0.05$, and $L = 40$.

emergence at $\chi = \chi_c$ and the stability over $\chi > \chi_c$ of the solution of the equation S1 and S2 Eqs in S2 Text for the plasmid surfing on the traveling wave (Fig 2B, red dots). Therefore, these results demonstrated the existence of solutions for plasmid surfing on the traveling wave of ParA-ATP for a 1D case.

Furthermore, we performed linear stability analysis of surfing-on-wave solution for S3 and S4 Eqs in S2 Text against external disturbances, in order to examine if tiny perturbation

around the stationary solution is amplified or not. We took $U(z, t) = U^{st}(z) + e^{\lambda t} \delta U(z)$ and $z_\xi(t) = z_\xi^{st} + e^{\lambda t} \delta z_\xi$, and computed eigenvalues $\lambda$ as a function of $\chi$ (S5 Fig and S2 Text). Then, the stability of the plasmid-surfing solution (Fig 2C, green) and the instability of the plasmid-localized solution (Fig 2C, purple) were given by the absence of $Re[\lambda(\nu)] > 0$, $\nu \neq 0$ and the existence of max $Re[\lambda(0)] > 0$, respectively (S5 Fig and S2 Text). If S16 Eq with $\nu \neq 0$ in S2 Text contained any positive real parameter $Re[\lambda(\nu)] > 0$, the perturbation around the traveling wave solution would be amplified to collapse; however, this is not the case. The results showed that plasmid surfing on the traveling wave emerges through a symmetry-breaking transition at a critical maximum rate of ParA-ATP hydrolysis ($= \chi_c$) as a pitchfork bifurcation in dynamical systems theory (Fig 2B and S5 Fig).

**Plasmid size is a relevant parameter for the emergence of its directed movement.** In the limit with vanishing plasmid size $l_b \to 0$, the mathematical form of the present model is reduced to our earlier model (S1 Text). However, whether any spontaneous directed motion emerges in the limiting case or not was not discussed in our previous report [43]. In such vanishing size limit, $\chi$ and $\varepsilon$ diverge: $\chi = kN/(2l_b) \to \infty$ and $\varepsilon = \mathcal{D}N/(2l_b) \to \infty$; nevertheless steady surfing-on-wave solutions can exist in a certain range of $k$, $\mathcal{D}$, and $N$. By using the analytical solution obtained above (S7-S9 Eqs in S2 Text) and by taking a large system size limit $L \to \infty$, the steady velocity in the limit is analytically obtained as (S25 Eq in S2 Text):
$\nu = \pm\sqrt{[kN(\mathcal{D}N/2 + 1) + 2][kN(\mathcal{D}N/2 + 1) - 2]}$. This analytical representation has a bifurcation at $k = k_c := 2/[N(\mathcal{D}N/2 + 1)]$: for $k$ larger than this critical value, there exist two real-valued solutions representing the directed movement. Corresponding to Fig 2A and 2B, we obtained the phase diagram and the relationship between $|\nu|$ and $k$ for $\mathcal{D} = 0.05$ and $N = 40$, for the parameter values adopted in Figs 1 and 2 (S6 Fig). The bifurcation point $k_c(= 0.025)$ in the vanishing size limit (S6B Fig) is smaller than the corresponding value ($k_c \sim 0.031$) for $l_b = 0.2$ in Fig 2B, suggesting that the directed motion for a larger $l_b$ requires a larger $k$, and probably $N$ as well.

We also examined $l_b$ dependency of the directed motion of the plasmid for the cases $N = 10$, 20, 30, 40, 50, using the analytical solution S7-S9 Eqs in S2 Text (Fig 2D). There were no solutions showing the directed movement for $N = 10$, whereas, for $N = 20, 30, 40, 50$, there exists a solution with the directed movement, whose velocity monotonously decreased with $l_b$, and then an inverse pitchfork bifurcation occurred at a critical value of $l_b$, resulting in the only solution with $\nu = 0$ (Fig 2D). Next, we numerically calculated $l_b$ dependency of the velocity and confirmed the disappearance of the traveling wave solution with the increase in $l_b$ (Fig 2D, black dot plots). The numerical results showed slight deviation from the analytical solution for the ranges of small and large $l_b$, suggesting the breakdown of the approximation $u(x) \gg K_d$. In general, the directed motion with larger $l_b$ requires a larger $k$ and $N$. This is because a plasmid (or bead) with large $l_b$ cannot generate sufficient concentration gradient and the resultant chemophoresis force is not sufficient to realize its self-driven directed motion. Finally, these results indicate that plasmid size is a relevant parameter for the emergence of its directed movement. Actually, in a previous mathematical study [37], its size relative to cell length was reported to play a different role in plasmid partition than the present result.

## Chemophoresis engine can work in *in vivo* conditions

From the normalized equations for chemphoresis engine (Eqs 3 and 4), we here confirmed that plasmid surfing on the traveling wave emerged through a symmetry-breaking transition with the change of $\chi$ and $\varepsilon$ (Fig 2). However, it remains unclear whether the above results can be quantitatively reproduced for the parameter values reported experimentally in bacterial cells. To quantitatively confirm that the directed motion is self-organized by the

chemophoresis engine, we ran the simulations again with parameters capturing *in vivo* conditions (S1 Table), as described below.

A recent study suggests that ParB localizes around plasmids by a ParA-ATP dependent phase separation mechanism to form PCs as droplets, accompanied by an enhanced hydrolysis activity of ParA-ATP through the increase in local density of ParB within the PCs, finally leading to a successful plasmid partition [75, 76]. As a simple demonstration, we carried out the simulation by simultaneously changing the maximum rate of ParA-ATP hydrolysis $\chi = kN/(2l_b)$ and the strength of the chemophoresis force $\varepsilon = \mathcal{D}N/(2l_b)$ under the control of the local density of ParB on a PC, $\rho := N/(2l_b)$. We increased the cell length, which may reflect the cell cycle progression, whereas we adopted the parameters capturing *in vivo* conditions listed in S1 Table. We changed $\rho$ and the cell length $= L \times l$ with $L$ as a normalized system size, scaled by $l = \sqrt{D_u \tau} = \sqrt{D_u/a} = 0.4(\mu m)$, while $\rho$ is a dimensionless relative density normalized by the cytoplasmic one of ParA-ATP assigned to $u_0$ (See also S1 Text for normalized parameters and S1 Table for *in vivo* parameter values).

We have performed simulations for the single plasmid ($M = 1$, S7 Fig upper) and the two-plasmid ($M = 2$, S7 Fig bottom) cases using Neumann boundary condition as in Fig 1. We investigated how the directed (or pole-to-pole oscillatory) movement of plasmids emerges with the increase in $\rho$, by calculating time-averaged values of velocity for different cell lengths (S7A Fig). The directed motion appeared at a $\rho = \rho_c$ for cell lengths greater than $0.8(\mu m)$ for $M = 1$ (S7A1 Fig) and $1.6(\mu m)$ for $M = 2$ (S7A2 Fig), demonstrating that the chemophoresis engine can recapitulate plasmid surfing even in *in vivo* conditions. Interestingly, the bifurcation points differ by different cell lengths; that is, the emergence of directed movement can be cell-length dependent. This bifurcation to the directed motion against the cell length occurs for both $M = 1$ and $M = 2$ (S7B1 and S7B2 Fig). In addition, the bifurcation diagram for $M = 1$ coincides with that for $M = 2$ when plotted as a function of (cell-length)/$M$ for the both cases (S7B Fig), suggesting that controlling the inter-plasmid distance is important for the emergence of directed motion. Such a scaling for bifurcation diagram among different values of $M$, as was mentioned in [37], can enable robust equi-positioning of plasmids during cell elongation [16, 67].

## Discussion

In this study, to consider a generalized model of the plasmid partition ParABS system, a chemophoresis engine was introduced as a coupled dynamical system among the equations of motion for plasmids and the RD equation for ParA-ATP (Fig 1A). In the model, plasmid dynamics switched from static to dynamic mode with an increase in the maximum rate of ATP hydrolysis $\chi$. The engine demonstrated equi-positioning, directed movement, and pole-to-pole oscillation, as observed in bacterial cells and *in vitro* experiments (Figs 1C, S1 and S2). Note that despite the plasmids' oscillatory behavior, the regular positioning distributions were sustained (S1 and S2 Figs) due to an effective inter-plasmid repulsive interaction derived from the chemophoresis force, indicating the robustness of positional information generated by the chemophoresis engine. By simplifying Eqs 3 and 4, and introducing a space-time coordinate, we mathematically showed the existence (Fig 2B and 2C) and the stability (S5 Fig) of the plasmid-surfing pattern. The solution emerged through the symmetry-breaking transition of the ParA-ATP spatiotemporal pattern at a critical $\chi$. We mathematically showed the directed movement emerges even in the limiting case of vanished plasmid-size $l_b \to 0$ (S6 Fig). Also, with an increase of the plasmid size $l_b$, the solution for the directed movement disappeared as a result of an inverse pitchfork bifurcation (Fig 2D). By using parameters capturing *in vivo*

conditions, we demonstrated that the chemophoresis engine can work even in bacterial cells (S7 Fig).

The plasmid surfing also worked for a two-dimensional (2D) case (S8 and S9 Figs). The simulation results for the 2D case in Eqs 3 and 4 are shown for $\chi = 10$ (S8A Fig) and $\chi = 50$ (S8B Fig). In the former case, the cargo maintained its location, and $u(r)$ had a symmetrical shape (S9A Fig), whereas directed motion by surfing on an asymmetrical traveling wave of $u(r)$ was observed for the latter (S9B Fig), just like the 1D case.

Although we analyzed the existence and stability of the surfing-on-wave pattern only in a noiseless situation (Fig 2), plasmids (or cargos) are always subjected to thermal fluctuations in cellular environments. Then, the plasmid motion was described by Langevin equation Eq 4. Further, we needed to elucidate that the plasmid-surfing-on-traveling-wave pattern remains robust against thermal fluctuations. For the chemophoresis force to act effectively, the force must be larger than the thermal noise, as discussed in a previous report [43]. Any force weaker than thermal noise cannot sustain even regular positioning [43]. We also examined how equi-positioning of plasmids can overcome thermal noise disturbances in a 1D case (S1 and S2 Figs). We confirmed that plasmid location dynamics shows a transition from stochastic switching to (freezing) steady equi-positioning (S1 and S2 Figs) as $\chi$ is increased, finally leading to persistent directed motion of the plasmids. This result suggested that the chemophoresis force dominates and directed movement of plasmids can overcome against thermal fluctuation for large hydrolysis rate.

We propose a chemophoresis engine, a general mechano-chemical apparatus driving the self-motion of the intracellular cargo, as a means to elaborate the physical principles of ATPase-driven cargo transport [48–53]. The engine is based on 1) a chemophoresis force that allows motion along an increasing ATPase(-ATP) concentration and 2) an enhanced catalytic ATPase hydrolysis at the cargo positions. ATPase-ATP molecules are used as fuel to supply free energy by applying the chemophoresis force along the concentration gradient, whereas cargos generate a concentration gradient by catalyzing the hydrolysis reaction on their surface. Note that each cargo, as a catalyst, does not consume ATP, but only modulates the concentration pattern. Through the coupling and synergy between 1) and 2), directed movement of the cargo is self-organized, showing a "surfing-on-traveling-wave" pattern (Fig 2C). The chemophoresis engine is based only on these two general mechanisms and is expected to explain how the transportation of diverse cargos in bacterial and eukaryotic cells is organized. Although we have focused on the gradient generated by the regulation of ATPase, the regulation of the concentration gradient via phosphorylation-dephosphorylation reactions is ubiquitous. Therefore, the chemophoresis engine resulting from the regulation of the hydrolysis of other factors, such as GTPase, should work for a variety of intracellular processes [77–80].

Our theory is derived from macroscopic thermodynamics under nonequilibrium conditions, and although we have applied it here to the ParABS system, it is general enough to be independent of individual microscopic models constructed for each molecular mechanism. In the present study, the mathematical model of the chemophoresis engine is constructed only by extracting the essential parts of the phenomena, so it can be applied directly to other systems with common reaction mechanisms such as hydrolysis. Indeed, it has been reported in *in vitro* experiments that the directed motion of a micro-sized bead is self-driven by the RNA gradient which RNA hydrolysis on the bead generates [81]; The authors later termed this phenomenon "autochemophoresis" [82].

Furthermore, recent studies demonstrated substrate-driven chemotactic behaviors of metabolic enzymes (single-molecule chemotaxis) [83, 84], and discussed a mathematical model to recapitulate experimental results in the subsequent study [85]. Interestingly, the authors proposed the exact same thermodynamic mechanism as the chemophoresis force described

previously [42, 43]. Therefore, we expect that the chemophoresis engine can also be applied to self-chemotactic behaviors even at a single-molecule level even though in the present study, self-chemotaxis is applied to a cargo size ranging from 50 nm to $1\mu$m. However, to describe nanoscopic chemotaxis, we need to extend thermodynamics of chemophoresis to a stochastic one which is valid even under large thermal/chemical fluctuations.

The merits of the chemophoresis engine are as follows: Self-generated chemical gradient for the chemophoresis force to apply; not requiring a large space to maintain the external chemical gradient. The chemophoresis engine can be effective in a moderate space. Therefore, the chemophoresis engine would work for eukaryotic intra-nuclear processes by restricting the mobility of chemicals on a nuclear membrane or a nuclear matrix functioning as a scaffold matrix. Based on the generality of the chemophoresis engine as well as suggestive reports in other systems [81, 82, 86], we can apply the mechanism to other hydrolysis events, RNAs, receptors, and others. We propose the chemophoresis engine as a general mechanism for hydrolysis-driven cargo transports in cells.

## Methods

### Numerical methods for solving evolutionary equation, self-consistent equation, and eigenvalue equation

Evolutionary equations, Eqs 3 and 4 were computationally solved as a hybrid simulation between reaction-diffusion equation and Langevin equation. Euler scheme for Eq 3 and Euler-Maruyama scheme for Eq 4 were used as numerical algorithms. Real-valued self-consistent equation, S9 Eq in S2 Text and complex-valued eigenvalue equation S16 Eq in S2 Text were solved by using Newton-Raphson method.

Since ParA-ATP always exists as a dimer on a nucleoid, it is reasonable to simply consider a hydrolysis reaction without any cooperativity in a spatially limited space around a plasmid. Therefore, the Hill coefficient $m$ was fixed as $m = 1$ in all the simulations. A normalized radius of plasmid $l_b$ was assigned to $l_b = 0.2$ through the simulation except for Fig 2D.

## Supporting information

**S1 Text. Details of the derivation of the chemophoresis engine.**
(PDF)

**S2 Text. Analytical solution of simplified equations.**
(PDF)

**S1 Fig. Plasmid location dynamics for M = 2.** The dynamics change among stochastic switching, steady equi-positioning, and directed movement followed by oscillatory mode as $\chi$ increases among $\chi = 0.5$ **(A)**, $\chi = 2.5$ **(B)**, and $\chi = 10$ **(C)** (two inner figures). The corresponding ParA-ATP pattern dynamics also change among stochastic switching, steady equi-positioning, and oscillatory waves (left). The oscillatory behavior of plasmids does not disrupt time-averaged equi-positioning. Steady multi-modal distributions of plasmids are sustained (Compare **(B)** right and **(C)** right). $K_d = 0.1$, $\varepsilon = 5$, and $L = 5$. The distributions (right) were generated using $10^7$ samples over $10^5$ time step.
(PDF)

**S2 Fig. Plasmid location dynamics for M = 3.** The dynamics change among stochastic switching, steady equi-positioning, and directed movement followed by oscillatory mode as $\chi$ increases among $\chi = 0.5$ **(A)**, $\chi = 2.5$ **(B)**, and $\chi = 10$**(C)** (two inner figures). The corresponding ParA-ATP pattern dynamics also change among stochastic switching, steady equi-positioning,

and oscillatory waves (left). The oscillatory behavior of plasmids does not disrupt time-averaged equi-positioning. Steady multi-modal distributions of plasmids are sustained (Compare **(B)** right and **(C)** right). $K_d = 0.1$, $\varepsilon = 5$, and $L = 5$. The distributions (right) were generated using $10^7$ samples over $10^5$ time step.
(PDF)

**S3 Fig. Simulation results of Eqs 3 and 4.** $\chi = 2.5$ **(A)** and $\chi = 10$ **(B)** with $K_d = 0.001$. The red lines show the spatial pattern of $u(x, t)$ (=ParA-ATP) for each $t$. Green dots show a plasmid location for each $t$. In the former case **(A)**, the plasmid maintains its location, whereas it surfs on the traveling wave of $u(x, t)$ and moves unidirectionally in the latter **(B)**. These results were compared with the analytical results in the main text and in Fig 2. $\varepsilon = 5$ and $L = 40$.
(PDF)

**S4 Fig. Analytical solution of $\varepsilon\Delta\mu(v, \chi) - v$ for visualization of the self-consistent equation S9 Eq in S2 Text.** S9 Eq in S2 Text shows a pitch-fork bifurcation at $\chi = \chi_c \sim 3.1$, and has three solutions for $\chi > \chi_c$. $\varepsilon = 5$ and $L = 40$.
(PDF)

**S5 Fig. Linear stability analysis of the surfing-on-wave pattern $U^{st}(z)$ and $z_\xi^{st}(=0)$ against external disturbances.** From tiny perturbations around the stationary solution $U(z, t) = U^{st}(z) + e^{\lambda t}\delta U(z)$, and $z_\xi(t) = e^{\lambda t}\delta z_\xi$, the eigenvalues $\lambda$ were computed as a function of $\chi$. The stability of the plasmid-surfing pattern (blue line) and the instability of the plasmid-localized solution (red line) was confirmed for $\chi > \chi_c$ in the absence of any positive real parameters of $\lambda$, Re[$\lambda$ (v)] > 0, $v \neq 0$. It seems to regain stability at $\chi$ larger than $\chi \sim 5.9$. However, such localized solution cannot be numerically realized for $\chi > 5.5$ because $U^{st}(0)$ becomes negative and the solution is unphysical at $\chi \sim 5.5$ (inset) as a result of breaking the approximation $u(x) \gg K_d$. Therefore, the stable localized solution does not exist for $\chi \gtrsim 5.5$.
(PDF)

**S6 Fig. Phase diagram and bifurcation of $v$ for S25 Eq in S2 Text. (A)** Steady velocity ($v$) profile of plasmid movement for $0 < k < 0.1$ and $0 < \mathcal{D} < 0.1$ for $N = 40$. **(B)** Relationship between $v$ and $k$ in analytical solutions for S25 Eq in S2 Text. Solutions for directed movement ($|v| > 0$) emerge at $k = k_c = 0.025$ as a result of a supercritical pitchfork bifurcation, whereas a solution for localization ($v = 0$) exists over $0 \leq k \leq 0.1$ in $\mathcal{D} = 0.05$, and $N = 40$.
(PDF)

**S7 Fig. Chemophoresis engine can work in *in vivo* conditions. (A)** Relationship between time-averaged velocity $|v|$ and $\rho$ at cell lengths with 0.8, 1.0, 1.25, 1.5, 1.75, and 2.0 for a single-plasmid case $M = 1$ **(A1)**, and 1.6, 2.0, 2.5, 3.0, 3.5, and 4.0 for a two-plasmid case $M = 2$ **(A2)**. **(B)** Relationship between time-averaged velocity $|v|$ and the cell length at $\rho = 14$, 14.5, 15, 16, 17, and 18 for both $M = 1$ **(B1)** and $M = 2$ **(B2)**. See S1 Table for the other model parameters.
(PDF)

**S8 Fig. Simulation results for a two-dimensional (2D) case of Eqs 3 and 4.** Successive snapshots are shown for $\chi = 10$ **(A)** and $\chi = 50$ **(B)**. In the former case **(A)**, the plasmid (red circle) maintains its location at a minimum symmetrical shape of $u(r, t)$ (=ParA-ATP, green scale), whereas it moves unidirectionally with an asymmetrical pattern of $u(r, t)$ in the latter **(B)**. $K_d = 0.001$, $l_b = 0.2$, $\varepsilon = 5$, and $L^2 = 10 \times 10$.
(PDF)

**S9 Fig. Simulation results for a two-dimensional (2D) case of Eqs 2 and 3.** Snapshots are shown for $\chi = 10$ **(A)** and $\chi = 50$ **(B)**. In the former case **(A)**, $u(r, t)$ (=ParA-ATP, green scale)

is symmetrical, and the plasmid (red circle) is located at its minimum. In contrast, for the latter **(B)**, the symmetry of $u(\mathbf{r}, t)$ is broken, indicative of a traveling wave. The minimum of the asymmetrical $u(\mathbf{r}, t)$ is positioned at a location shifted from where the plasmid lies, suggesting that the plasmid in "surfing" on the traveling wave." $K_d = 0.001$, $l_b = 0.2$, $\varepsilon = 5$, and $L^2 = 10 \times 10$.
(PDF)

**S1 Table. Model parameters capturing *in vivo* conditions.**
(PDF)

## Acknowledgments

We thank Hironori Niki, Kazuhiro Maeshima, Akatsuki Kimura, Hiraku Nishimori, Akinori Awazu, Satoshi Sawai, Nobuhiko J. Suematsu, Satoshi Nakata, Sosuke Ito, and Yasushi Okada for their comments.

## Author Contributions

**Conceptualization:** Takeshi Sugawara, Kunihiko Kaneko.

**Formal analysis:** Takeshi Sugawara.

**Funding acquisition:** Takeshi Sugawara, Kunihiko Kaneko.

**Investigation:** Takeshi Sugawara.

**Methodology:** Takeshi Sugawara.

**Project administration:** Takeshi Sugawara, Kunihiko Kaneko.

**Writing – original draft:** Takeshi Sugawara, Kunihiko Kaneko.

**Writing – review & editing:** Takeshi Sugawara, Kunihiko Kaneko.

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
