## [Decision Letter · Decision Letter 0]

18 Feb 2022

Dear Dr. Sugawara,

Thank you very much for submitting your manuscript "Chemophoresis Engine: Universal Principle of ATPase-driven Cargo Transport" for consideration at PLOS Computational Biology.

As with all papers reviewed by the journal, your manuscript was reviewed by members of the editorial board and by several independent reviewers. In light of the reviews (below this email), we would like to invite the resubmission of a significantly-revised version that takes into account the reviewers' comments.

We cannot make any decision about publication until we have seen the revised manuscript and your response to the reviewers' comments. Your revised manuscript is also likely to be sent to reviewers for further evaluation.

Sincerely,

Alexandre V. Morozov, Ph.D.

Associate Editor

PLOS Computational Biology

Daniel Beard

Deputy Editor

PLOS Computational Biology

Reviewer's Responses to Questions

**Comments to the Authors:**

Reviewer #1: Review attached separately in PDF file.

Reviewer #2: Sugawara and Kaneko published the first theory paper on Chemophoresis a decade ago. Since then, the mathematical modeling studies from other labs have greatly extended this concept and applied to plasmid partition along with other cargo positioning in bacteria in several systems, both in vivo and in vitro. In this manuscript, Sugawara and Kaneko formulated the mathematical framework of chemophoresis, perhaps more vigorously. Again, they applied the model to ParA-mediated plasmid partition and recapitulated the different patterns such as plasmid oscillation and equi-distancing, which have been explained by other models as the authors cited. Overall, the theoretical work is solid and potentially transformative, in particular, in light of a general mechanism for cellular positioning. This work points out a potentially new mechanism of cargo positioning inside cell and thus holds a great interest in the field of cell motility and theory.

However, I fail to see what the new thing is in the current version as compared to their published work in Biophysics, nor how the work on plasmid partition can be extended to other systems as a general mechanism. I feel that the work as it currently stands may be more suitable for a specialized journal. On the other hand, the writing of this draft may mask something important. For instance, did the authors derive a new analytical formula/a more general mathematical model to predict different mobility patterns & delineate the conditions that cell can use this mechanism for transport and position cargos? If so, I’d suggest the authors to 1) make these points very explicit, and 2) discuss what can be explained by this work but not by others. And I would be more than happy to review it again.

**Have the authors made all data and (if applicable) computational code underlying the findings in their manuscript fully available?**

Reviewer #1: None

Reviewer #2: Yes

PLOS authors have the option to publish the peer review history of their article (what does this mean?). If published, this will include your full peer review and any attached files.

Reviewer #1: No

Reviewer #2: No
---

## [Decision Letter · Decision Letter 1]

8 Jun 2022

Dear Dr. Sugawara,

Thank you very much for submitting your manuscript "Chemophoresis Engine: a General Mechanism of ATPase-driven Cargo Transport" for consideration at PLOS Computational Biology. As with all papers reviewed by the journal, your manuscript was reviewed by members of the editorial board and by several independent reviewers. The reviewers appreciated the attention to an important topic. Based on the reviews, we are likely to accept this manuscript for publication, providing that you modify the manuscript according to the review recommendations.

Sincerely,

Alexandre V. Morozov, Ph.D.

Associate Editor

PLOS Computational Biology

Daniel Beard

Deputy Editor

PLOS Computational Biology

[LINK]

Reviewer's Responses to Questions

**Comments to the Authors:**

Reviewer #1: My referee comments have been adequately addressed in the revised manuscript. Especially, my concerns about the parameters in relation with real measurements are well addressed.

Some claim of the paper is a bit hypothetical but still interesting to read. I think it is worthwhile for publication in PLOS Comp. Biol.

Reviewer #2: The authors addressed my comments and emphatically clarified their new findings. I'd only suggest one change: The authors claimed that "Plasmid size and cell length dependence has not been reported in the previous mathematical studies." This is not true. The new paper by Hu et al (ref #38) addressed the PC size scaling. It would make sense that the authors modify the relevant paragraphs (e.g., lines 239-268) to make connection to the findings in ref #38 and to parse out the commonality and differences.

**Have the authors made all data and (if applicable) computational code underlying the findings in their manuscript fully available?**

Reviewer #1: None

Reviewer #2: Yes

PLOS authors have the option to publish the peer review history of their article (what does this mean?). If published, this will include your full peer review and any attached files.

Reviewer #1: No

Reviewer #2: No

Figure Files:

Data Requirements:

Reproducibility:

References:

---

## [Editor Report · Decision Letter 2]

23 Jun 2022

Dear Dr. Sugawara,

We are pleased to inform you that your manuscript 'Chemophoresis Engine: a General Mechanism of ATPase-driven Cargo Transport' has been provisionally accepted for publication in PLOS Computational Biology.

Best regards,

Alexandre V. Morozov, Ph.D.

Associate Editor

PLOS Computational Biology

Daniel Beard

Deputy Editor

PLOS Computational Biology

---

## [Editor Report · Acceptance letter]

15 Jul 2022

PCOMPBIOL-D-21-02109R2 

Chemophoresis Engine: a General Mechanism of ATPase-driven Cargo Transport

Dear Dr Sugawara,

I am pleased to inform you that your manuscript has been formally accepted for publication in PLOS Computational Biology. Your manuscript is now with our production department and you will be notified of the publication date in due course.

With kind regards,

Olena Szabo
